behaviour/ecology/environmental science

brood size, costs-benefits, defending behaviour, disease transmission, prey spectrum, urbanization

**Author for correspondence:**
Manuela Merling de Chapa
e-mail: manu.merling@googlemail.com

# Phantom of the forest or successful citizen? Analysing how Northern Goshawks (*Accipiter gentilis*) cope with the urban environment

Manuela Merling de Chapa[1], Alexandre Courtiol[1], Marc Engler[1], Lisa Giese[1], Christian Rutz[2], Michael Lakermann[3], Gerard Müskens[4], Youri van der Horst[5], Ronald Zollinger[6], Hans Wirth[7], Norbert Kenntner[3], Oliver Krüger[8], Nayden Chakarov[8], Anna-Katharina Müller[8], Volkher Looft[9], Thomas Grünkorn[10], André Hallau[3], Rainer Altenkamp[11] and Oliver Krone[1]

[1]Leibniz Institute for Zoo and Wildlife Research, Alfred-Kowalke-Straße 17, 10315 Berlin, Germany
[2]Centre for Biological Diversity, School of Biology, University of St Andrews, St Andrews KY16 9TH, UK
[3]Independent Researcher, Germany
[4]Wageningen Environmental Research (WENR), Animal Ecology Team, PO Box 47, NL-6700 AA, Wageningen, The Netherlands
[5]Vogeltrekstation, Dutch Centre for Avian Migration and Demography (NIOO-KNAW), Postbus 50, 6700 AB Wageningen, The Netherlands
[6]Natuurplaza, PO Box 1413, NL-6501 BK Nijmegen, The Netherlands
[7]Ornithologische Arbeitsgemeinschaft Schleswig-Holstein e.V., Wiesengrund 11, 22967 Tremsbüttel, Germany
[8]Department of Animal Behaviour, Bielefeld University, Morgenbreede 45, 33615 Bielefeld, Germany
[9]Landesnaturschutzverband Schleswig-Holstein e.V., Burgstraße 4, 24103 Kiel, Germany
[10]BioConsult SH, Schobüller Straße 36, 25813 Husum, Germany
[11]NABU Berlin, Wollankstraße 4, 13187 Berlin, Germany

MM, 0000-0003-2585-4275; AC, 0000-0003-0637-2959; CR, 0000-0001-5187-7417; OKro, 0000-0002-4507-5124

By 2040, roughly two-thirds of humanity are expected to live in urban areas. As cities expand, humans irreversibly transform natural ecosystems, creating both opportunities and challenges

for wildlife. Here, we investigate how the Northern Goshawk (*Accipiter gentilis*) is adjusting to urban environments. We measured a variety of behavioural and ecological parameters in three urban and four rural study sites. City life appeared related to all parameters we measured. Urban female goshawks were overall 21.7 ($CI_{95\%}$ 5.13–130) times more likely to defend their nestlings from humans than rural females. Urban goshawks were 3.64 ($CI_{95\%}$ 2.05–6.66) times more likely to feed on pigeons and had diets exhibiting lower overall species richness and diversity. Urban females laid eggs 12.5 ($CI_{95\%}$ 7.12–17.4) days earlier than rural individuals and were 2.22 ($CI_{95\%}$ 0.984–4.73) times more likely to produce a brood of more than three nestlings. Nonetheless, urban goshawks suffered more from infections with the parasite *Trichomonas gallinae*, which was the second most common cause of mortality (14.6%), after collisions with windows (33.1%). In conclusion, although city life is associated with significant risks, goshawks appear to thrive in some urban environments, most likely as a result of high local availability of profitable pigeon prey. We conclude that the Northern Goshawk can be classified as an urban exploiter in parts of its distribution.

## 1. Introduction

Urbanization constitutes one of the most dramatic human-driven transformations of natural ecosystems [1]. By 2040, approximately two-thirds of the world´s human population may be living in urbanized areas [2]. For wildlife, urbanization can present both opportunities and major challenges. While some species are extremely sensitive to habitat disturbance and disappear quickly, others can adapt to urban habitats while continuing to use natural resources, and some species even thrive as urban commensals, to the point that they become dependent on urban resources [3]. Accordingly, Blair [4] proposed to classify species into 'urban avoiders', 'suburban adapters' and 'urban exploiters', depending on how they cope with the urban environment.

The Northern Goshawk (*Accipiter gentilis*) (henceforth 'goshawk') is sometimes referred to as the 'phantom of the forest' by bird watchers and raptor enthusiasts, because of its secretive habits. In the past, persistent persecution by humans presumably favoured relatively shy individuals that avoid humans [5,6]. Goshawks are thus considered urban avoiders throughout most of their range [7]. Nevertheless, goshawks have started colonizing several European cities [6], notably in Germany, where first urban breeding attempts occurred in the 1980s, followed by rapid population expansions in the 1990s, and saturation in recent years [8–10]. In these cities, the species has reached some of the densest populations recorded for the species worldwide [8–12].

Density alone, however, is not an adequate measure of successful colonization. Perceived habitat attractiveness could result in high population densities in cities, even though urban individuals may experience higher mortality and/or reduced fecundity rates compared with rural areas, resulting in an ecological trap [13]. For example, Cooper's hawks in Arizona are attracted to cities by vacant nesting sites and abundant prey, leading to stable populations; yet, breeding pairs suffer from rates of parasite-induced nestling mortality that are so high that urban populations would not persist if it were not for the constant immigration of individuals from surrounding rural areas [14,15]. Similar population trajectories are also conceivable for goshawks. More generally, understanding why some species adjust well to urban habitats and the presence of humans, while others do not [16,17], is a major research challenge in urban ecology. Comprehensive studies encompassing measures of both potential benefits and costs of living in urban habitats are hence urgently needed.

Here, we investigate, with appropriate population-level replication, how goshawks have adjusted to urban habitats. We start by analysing the potential benefits of urban life, by measuring three different variables: (i) *behavioural responses*, as behaviour is a key determinant for the success of wildlife in urban environments [17,18], (ii) *diet composition*, since urban goshawks appear to specialize on hunting pigeons, which are particularly abundant in cities [12,19,20], and (iii) *breeding performance*, as some studies reported that goshawks start breeding earlier in cities and/or exhibit higher fecundity rates in urban than in rural areas [9,10,20]. We then assess two variables that could reveal potential costs of living in urban habitats: (iv) *health status*, which could be poorer in urban goshawks if their diet contains more pigeons, which are the main host of *Trichomonas gallinae*—the agent of the disease trichomonosis [21,22], and (v) *causes of mortality*, since the pursuit hunting style of goshawks could make it particularly vulnerable to collisions with human-made obstacles in urban areas [23–25].

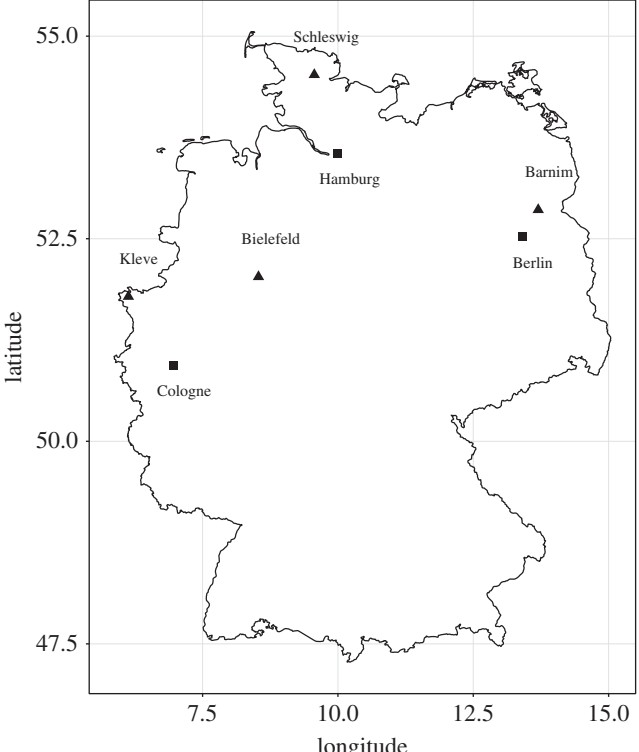

**Figure 1.** Map of the locations of the seven study populations. Squares represent urban sampling locations and triangles rural ones.

# 2. Material and methods

## 2.1. Study sites and samples

The study was carried out between 2014 and 2016 in three urban and four rural study sites in northern parts of Germany (figure 1). The urban sites were three cities with human populations of over 1 million each: Berlin, Cologne and Hamburg. The rural 'control' sites were located in the area of Barnim, the lower Rhine area near Kleve, rural areas around Schleswig, as well as areas inside the Teutoburg forest around Bielefeld.

Goshawk territories were surveyed within each of the seven locations, resulting in the monitoring of 20 to 60 nests per location per year. Each nest was checked every two to four weeks for signs of successful breeding from February to June. Nestlings in successful nests were banded and sampled at a minimum age of approximately 14 days. Nestlings were handled on the ground and later returned to the nest. Some territories were investigated over multiple study years. In total, we were able to sample 544 goshawk nestlings in 196 nests in 133 different territories (table 1). The age of breeding goshawks was classified as first year, second year and adult (≥ three years) based on plumage colour patterns [5]. From the 196 sampled nests, we were able to identify the age of the breeding female in 116 cases. All but two individuals were identified as adults. For this reason, we excluded the variable 'age of breeding goshawks' from further analyses.

All procedures were performed in accordance with the requirements of the Leibniz Institute for Zoo and Wildlife Research Ethics Committee on Animal Welfare. The banding and handling of goshawk nestlings was approved by the ornithological stations of Radolfzell, Hiddensee and Helgoland.

## 2.2. Measurements

### 2.2.1. Behavioural responses

Female goshawks tend to remain at, and defend, the nest throughout the nestling period [5]. We measured females' reaction to the person climbing the nest tree at the time of banding nestlings in 2015 and 2016, distinguishing four response levels: 0 = no reaction, 1 = alarm calls, 2 = feint attack and 3 = physical attack. An aggression level of 0 can include cases when the bird was absent (after three weeks, adult females may leave the nest site to hunt), or when it was present at the nest site but was not seen and did

**Table 1.** Sample sizes for nestlings (and territories) per study location and year.

| habitat | location | 2014 | 2015 | 2016 | total per location | total per habitat |
|---|---|---|---|---|---|---|
| urban | Berlin | 61 (19) | 58 (19) | 59 (19) | 178 (57) | 285 (96) |
| | Cologne | — | 36 (12) | 26 (9) | 62 (21) | |
| | Hamburg | — | 17 (8) | 28 (10) | 45 (18) | |
| rural | Bielefeld | 13 (6) | 15 (6) | 14 (7) | 42 (19) | 259 (100) |
| | Barnim | 21 (7) | 21 (8) | — | 42 (15) | |
| | Kleve | 11 (4) | 42 (18) | 60 (20) | 113 (42) | |
| | Schleswig | 19 (7) | 16 (8) | 27 (9) | 62 (23) | |

not react; we were unable to distinguish between these scenarios, highlighting the scope for systematic sampling bias [26]. Nestlings were of similar age in urban and rural areas at the time of sampling (median age of youngest nestlings for urban territories = 21 days, $N$ = 77; rural territories = 20 days, $N$ = 74; Mann–Whitney U test: $W$ = 4315.5, $p$ = 0.268). Nevertheless, since the availability of prey may differ between the two habitats, it is possible that parents spent more time foraging and thus were more often absent from the nest area, in the habitat exhibiting comparatively lower prey availability. We therefore distinguished nests with young nestlings (16 days and younger) from those with older nestlings in analyses (dummy coded); no major difference in female presence would be expected for young nestlings as they are strictly dependent on their mothers [5,27]. The reaction of males was not considered in our analyses, as they are routinely hunting outside the nest area during the nestling period [5].

### 2.2.2. Diet composition

Prey remains were collected approximately every two weeks during the breeding season in 2016 (from March until July), for the following number of nesting territories: 17 territories in Berlin; 14 in Hamburg; 27 in Cologne; 8 in the rural habitat near Barnim; 20 near Kleve; 7 near Schleswig and 0 near Bielefeld. Estimation of diet composition from prey remains was carried out following Rutz [28]. We also collected pellets and identified their prey content via genetic analysis (see electronic supplementary material). Pellets can reveal the presence of small mammals and birds in the diet of raptors which are otherwise difficult to record. To avoid double-counting of prey items on any given sampling day, we excluded species from pellets that were also present in prey remains. We calculated *species richness* as the number of different prey species recorded per territory, *overall species richness* as the total number of different prey species recorded per habitat, and *diversity* using Simpson's index [29]. We also constructed rarefaction curves to explore how species richness is affected by the number of territories sampled using the R package vegan v. 2.5–6 [30].

### 2.2.3. Breeding performance

To compare the breeding performance of urban and rural goshawks, we used the *laying date of the first egg* and the *brood size* of each adult female. Nestling age was determined by measuring wing length and body mass, following Bijlsma [31]. From this, we back-calculated the *laying date* as being 38 days (it varies between 35 to 43 days [5,32]) before the inferred hatching date. In statistical analyses, the laying date was coded as the number of days since 31 December (i.e. 1 January = 1, 2 January = 2 and so on). We defined the *brood size* as the number of nestlings present in the nest at the time of banding. We used a Mann–Whitney U test, as implemented by the function wilcox.test in R, to compare the brood size of rural and urban goshawks (without controlling for any confounding variables). For fitting linear models, we recoded brood size as a two-level factor (small brood: 1–2 nestlings; large brood: 3–5 nestlings) due to the particular distribution of this variable (see electronic supplementary material for details).

### 2.2.4. Health status

Nestlings were examined during banding between May and June, when they were between 11 and 40 days old (median age for rural nestlings = 23 days, $N$ = 257; median age for urban nestlings = 24 days, $N$ = 285; two nestlings were not measured). The body mass of a nestling combined with its wing

length measurement allowed us to distinguish males and females, as females are significantly larger [31]. All nestlings were tested for the presence of the endoparasite *T. gallinae* by taking a sample from the oesophagus using sterile cotton swabs and also examined for clinical signs of the disease trichomonosis. After sampling, swabs were stored in a special culture medium (InPouch TV, Biomed Diagnostics, White City, US) and placed in a mobile incubator at 37°C for an incubation period of 10 days. During this time, the presence of the flagellate parasite was checked every day, using a microscope. We considered samples to be negative when no parasite had been observed during the incubation period.

### 2.2.5. Causes of mortality

Causes of mortality were investigated for individuals found dead between 1998 and 2017. In total, 189 carcasses were collected (88 females and 101 males) by members of the general public, NGOs and nature conservation authorities. We determined that 151 birds were urban and 38 were rural based on the recovery location of the carcasses. We checked that sites did not change in their urbanization classification (urban versus rural) within the last few decades. We performed necropsies for 69 goshawks collected during the survey period. Out of these, 42 originated from the territories we followed. We were also able to age all but two carcasses based on plumage coloration [5] and the date of discovery: 7% were nestlings (collected in nests), 16.5% were juveniles (May to June), 46% were in their first year of life and 30.5% were adults. These numbers are consistent with the general observation that goshawks experience the highest mortality rates during their first year of life [5]. All necropsies were performed at the Leibniz Institute for Zoo and Wildlife Research, following the same protocol. Causes of mortality were inferred from pathologic findings along with information provided by the finder of the carcass.

## 2.3. Statistical analyses

All statistical analyses were performed using R v. 4.0.3 [33]. We analysed the influence of habitat type on our five focal variables by means of linear mixed-effects models (LMM), generalized linear mixed-effects models (GLMM) and generalized linear models (GLM) (tables 2 and 3). A justification for why we opted to use broad categories (urban versus rural) instead of a quantitative index, such as *imperviousness*, is given in the electronic supplementary material, together with detailed reports for all statistical models. Importantly, all models include a random effect structure that accounts for multiple measurements. All models were fitted using the package spaMM v. 3.5.32 [34]. We checked that the main assumptions of linear modelling (lack of serial autocorrelation, expected dispersion and distribution of residuals) were fulfilled using DHARMa v. 0.3.3 [35]. We computed the significance of fixed-effect parameters in all models using a likelihood ratio test (LRT): we compared the observed LRT statistic (hereafter $\chi^2$) to its distribution under the null hypothesis to compute the *p*-value. The latter distribution and the *p*-value were obtained through 1000 parametric bootstraps using the function *anova()* from spaMM. We computed 95% confidence intervals ($CI_{95\%}$) around some key parameter estimates using likelihood profiling in spaMM. Since likelihood profiling and parametric bootstrap are not strictly equivalent, small differences are expected and slightly significant *p*-values can sometimes correspond to a boundary of the $CI_{95\%}$ overlapping zero, or slightly non-significant *p*-values can sometimes correspond to boundaries excluding zero. In such cases, we relied on *p*-values obtained by parametric bootstrap to determine significance.

# 3. Results

## 3.1. Measurements

### 3.1.1. Behavioural responses

In 70 out of 151 records (46.4%), combining urban and rural observations, the female did not react to a person climbing the nest tree. Among the 81 remaining records, in 63 cases (41.7%), the female produced an alarm call; in 14 cases, she made a feint attack (9.3%), and in four cases involving the same two individuals across years, the climber was physically attacked (2.6%). Importantly, urban and rural individuals exhibited very different response rates (figure 2), with 66.2% (51 out of 77) of urban and only 16.2% (12 out of 74) of rural goshawks producing an alarm call, while 13.0% (10 out of 77) of urban and 81.1% (60 out of 74) of rural birds did not react.

**Table 2.** Summary of linear models implemented. The star symbol denotes an interaction.

| topic | response variable | type of the response | distribution | explanatory variables | random factors |
|---|---|---|---|---|---|
| behavioural responses | defending behaviour (GLMM) | binary data (reaction: 'yes' or 'no') | binomial | habitat*age categorical, year, number of nestlings, laying date, rainfall | locations, territories |
| diet composition | proportion of pigeons (GLMM) | binomial data (proportion of pigeons and other species per territory) | binomial | habitat | locations, territories |
| | species richness (GLMM) | count data (number of prey species per territory) | truncated negative binomial | habitat | locations[a] |
| | diversity (BoxCox transformed LMM) | continuous data (Simpson's Index of prey diversity per territory) | Gaussian | habitat | locations[a] |
| breeding performance | number of nestlings (GLMM) | binary data ('small brood' (1–2 nestlings) 'large brood' (3–5 nestlings)) | binomial | habitat, laying date, average temperature at breeding begin | locations, territories |
| | laying date (LMM) | continuous data (start of egg laying per territory) | Gaussian | habitat, average temperature at breeding begin | locations, territories |
| health status | prevalence of *T. gallinae* (GLMM) | binary data (infection: 'yes' or 'no') | binomial | habitat, year, age, sex, number of nestlings, laying date, average temperature during nestling age | locations, territories |
| | clinical signs of trichomonosis (GLMM) | binary data (clinical signs: 'yes' or 'no') | binomial | habitat, year, age, sex, number of nestlings, laying date, average temperature during nestling age | locations, territories |
| causes of mortality | trichomonosis (GLM) | binary data (cause of death: 'yes' or 'no') | binomial | habitat, sex, age | — |
| | window collision (GLM) | binary data (cause of death: 'yes' or 'no') | binomial | habitat, sex | — |

[a]prey remains were only sampled in 2016; it was therefore not necessary to fit territory as a random effect in these GLMs.

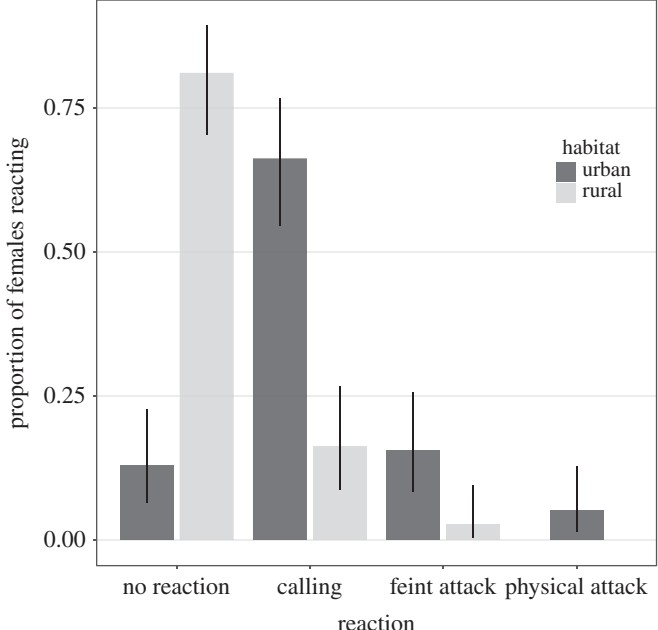

**Figure 2.** Proportion of different behaviours exhibited by female goshawks in response to our presence during nestling banding in urban and rural habitats, with $CI_{95\%}$.

**Table 3.** Description of explanatory variables.

| explanatory variable in statistical models | description | type of the explanatory variable |
|---|---|---|
| habitat | habitat types | qualitative (two categories: 'urban', 'rural') |
| year | years of sampling | qualitative (two to three categories: '2014', '2015', '2016') |
| age | age of the nestlings | quantitative (values from 11 to 40) |
| age qualitative | age class of nestlings (young = youngest nestling in the nest is 16 days or younger; old = youngest nestling in the nest is older than 16 days) | qualitative (two categories: 'young', 'old') |
| sex | sex of nestlings | qualitative (two categories: 'female', 'male') |
| number of nestlings | number of nestlings per single territory | quantitative (count from 1 to 5) |
| laying date | day first egg was laid using numeric values with one for 1 January and 365 for 31 December. | quantitative (values from 65 to 115) |
| average temperature at breeding begin | average temperature[a] per location from February to March of each year | quantitative (values from 3.5 to 5.5) |
| average temperature during nestling age | average temperature[a] from hatching day to sampling day per individual nestling | quantitative (values from 10.3 to 17.9) |
| rainfall | average amount of rainfall[a] (mm) at the sampling day | quantitative (values from 0 to 15.8) |

[a]data obtained from the 'Deutschen Wetterdienst', one average temperature value in °C or amount of rainfall value in mm per day and location.

This result was consistent with a GLMM analysis predicting the probability that females reacted to the climber. The strongest effect was indeed found for habitat type (GLMM, $n = 151$: $\chi^2 = 12.3$, $p = 0.016$): the odds that a female reacted were 10.8 times higher in the urban than in the rural habitat

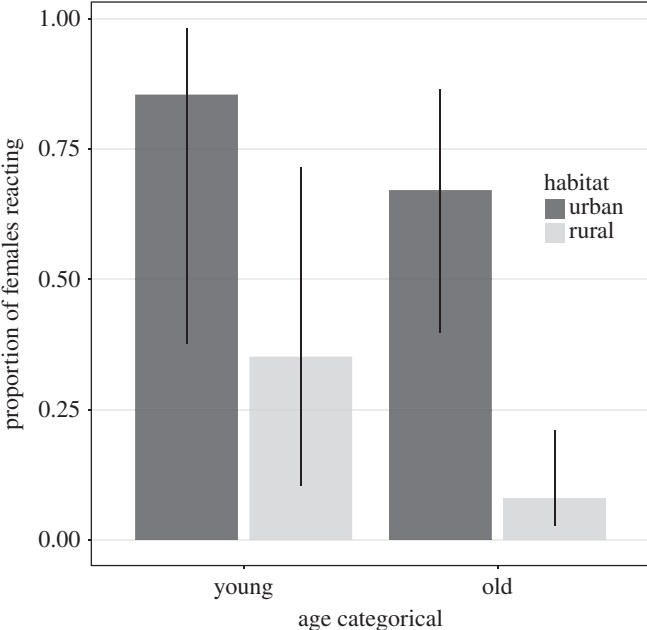

**Figure 3.** Predicted probability of reaction of female goshawks to our presence during nestling banding, depending on habitat and nestling age (young ≤ 16 days, old > 16 days). For all figures illustrating predictions from linear models (figures 3, 4, 7, 9 and 10), random effects were considered at zero in predictions, to show predictions for an average territory; covariates not illustrated were considered at their median value. Factors not illustrated were constrained (if applicable; see main text) as follows: number of nestlings = 3, age class = young, year = 2015, sex of nestlings = female, habitat = urban (dark grey) and rural (light grey). For this figure, and all others showing model predictions, we chose the year 2015, because all locations were sampled in this year. Dotted lines or error bars show $CI_{95\%}$.

when nestlings were less than 16 days old, and 23.5 times higher when they were older (figure 3). The effect of nestling age on the reaction of females was not clear. The odds that a female reacted were 2.87 times higher for young than for old nestlings in the urban habitat and 6.24 times in the rural habitat. Yet, our model does not support the presence of a strong interaction between age class and habitat type ($\chi^2 = 0.284$, $p = 0.736$); moreover, the overall difference between the two age classes was non-significant ($\chi^2 = 5.98$, $p = 0.083$). Nonetheless, the strong effect of habitat remained after removing the interaction between age class and habitat type from the model, with urban females being 21.7 times ($CI_{95\%}$ 5.13–130) more likely to react than rural ones ($\chi^2 = 12.0$, $p = 0.005$), irrespectively of the age of the nestlings ($\chi^2 = 5.70$, $p = 0.015$). Laying date also had a significant negative effect on goshawks' reaction ($\chi^2 = 4.94$, $p = 0.033$, figure 4), and behaviour differed between the two years analysed ($\chi^2 = 8.01$, $p = 0.011$), with the odds that a female reacted being 4.32 times higher in 2016 than in 2015. The other variables considered in the GLMM did not significantly influence the probability that females reacted to the climber's presence (rainfall: $\chi^2 = 0.699$, $p = 0.427$; number of nestlings: $\chi^2 = 1.04$, $p = 0.337$).

### 3.1.2. Diet composition

In total, we collected 888 prey items within 93 territories across six study populations: 546 in urban areas and 342 in rural areas (table 4); a complete prey list is provided in the electronic supplementary material, table S1.

Columbidae species represented 65.4% (355 out of 546) and 35.7% (122 out of 342) of all recorded prey items in urban and rural habitats, respectively. Feral pigeons and woodpigeons were the dominant prey species, followed by doves and other species (figure 5 and electronic supplementary material, table S1). On average, the odds for a prey item to be a pigeon rather than something else was 3.64 ($CI_{95\%}$ 2.05–6.66) times higher in urban than in rural habitats (GLMM, $n = 93$: $\chi^2 = 10.2$, $p = 0.015$).

In total, goshawks breeding in the rural habitat killed 39 different species, while urban goshawks only consumed 32 species despite the larger sample size for urban locations (electronic supplementary material, table S1). Accordingly, our fitted model predicted 1.44 fewer species per territory in urban areas (LMM, $n = 96$: $\chi^2 = 5.53$, $p = 0.018$), corresponding to a raw species richness per territory of 4.02 (range = 1–9, s.d. = 1.97, $n = 58$) and 5.46 (range = 1–13, s.d. = 3.38, $n = 35$) in urban and rural habitats,

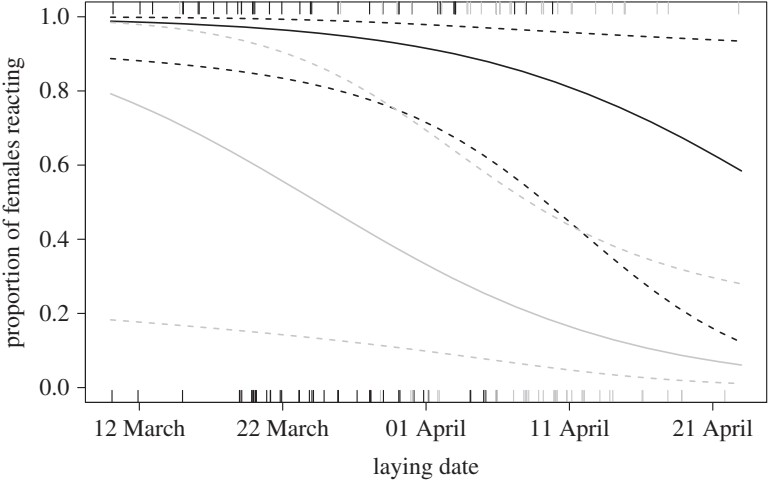

**Figure 4.** Predicted probability of reaction of female goshawks to our presence during nestling banding, depending on the laying date (the given dates illustrate a typical non-leap year). See legend of figure 3 for further details. Habitat = urban (dark grey) and rural (light grey). Raw data are shown as jittered ticks along the *x*-axis at the bottom (no reaction) or top (reaction) of the plot.

**Table 4.** Number of prey items found at each study location in 2016.

| habitat | location | number of territories | prey items | total number of prey items per habitat |
|---|---|---|---|---|
| urban | Berlin | 17 | 195 | 546 |
| | Cologne | 27 | 235 | |
| | Hamburg | 14 | 116 | |
| rural | Bielefeld | 0 | 0 | 342 |
| | Kleve | 20 | 208 | |
| | Barnim | 8 | 68 | |
| | Schleswig | 7 | 66 | |

respectively. The analysis of rarefaction curves confirms that the observed difference in species richness between urban and rural habitat is not caused by a difference in the sampling effort (electronic supplementary material, figure S2).

Simpson's diversity index differed significantly between habitats (BoxCox transformed GLMM, $n = 67$: $\chi^2 = 8.93$, $p = 0.008$), with diversity predicted to be 0.11 points lower for urban territories. This pattern was also obvious in the raw data, with an average Simpson's diversity index of 0.66 (range = 0.156–0.857, s.d. = 0.163, $n = 43$) for urban territories and of 0.79 (range = 0.283–0.903, s.d. = 0.133, $n = 24$) for rural ones.

### 3.1.3. Breeding performance

On average, *egg laying* in cities took place at day 83 (i.e. 25 March; in a typical non-leap year; range = 68–99, s.d. = 7.58, $n = 96$), compared with day 97 (i.e. 8 April) for rural habitats (range = 73–112, s.d. = 7.30, $n = 99$). According to the model, which controls for other differences between the two habitats, urban goshawks began egg laying 12.5 ($CI_{95\%}$ 7.12–17.4) days earlier, corresponding closely to the mean values for the raw data (14 days); this difference was significant (LMM, $n = 195$: habitat: $\chi^2 = 11.3$, $p = 0.009$), even after controlling for the effect of temperature ($\chi^2 = 1.61$, $p = 0.221$).

The mean *brood size* per territory was 2.85 (range = 1–5, s.d. = 0.923, $n = 195$) across all territories studied. We found an average of 3.06 nestlings per nest in urban areas (1–5, s.d. = 0.927, $n = 96$), compared with 2.63 nestlings per nest (1–4, s.d. = 0.872, $n = 100$) in rural ones (figure 6) (Mann–Whitney U test: $W = 3517.5$, $p < 0.001$). Dividing the broods into large ($\geq 3$ nestlings) and small ($\leq 2$ nestlings), to obtain a distribution suitable for statistical modelling (see electronic supplementary material), a similar pattern was found: 74.0% of broods were large in urban habitats (71 out of 96) but only 55.6% in rural habitats (55 out of 99). Yet, this difference did not reach significance in a GLMM

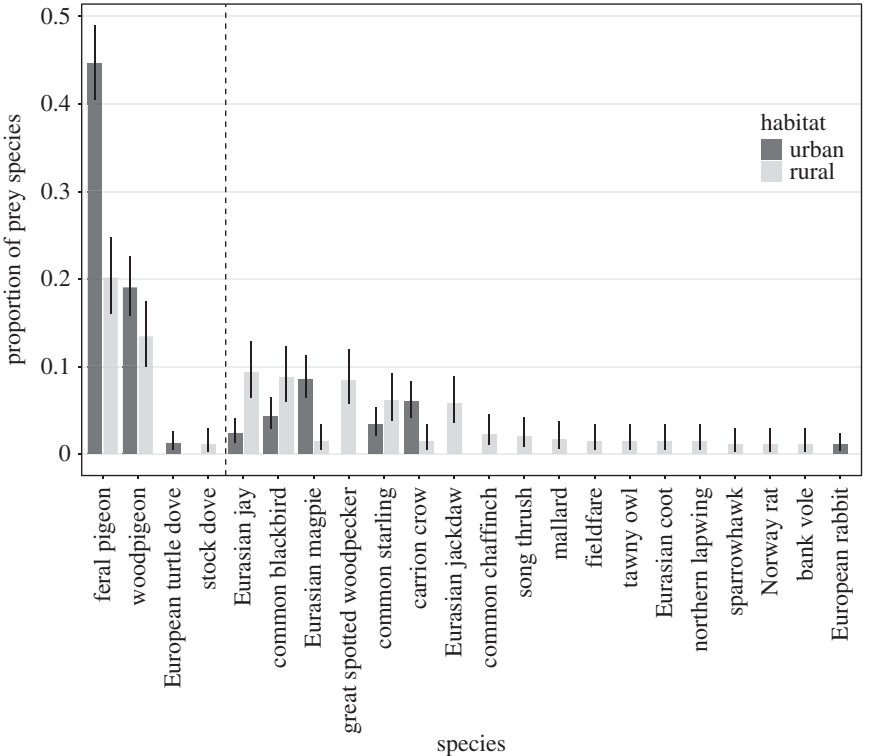

**Figure 5.** Proportion (with CI$_{95\%}$ the binomial probability) of species in the diet of urban and rural goshawks above 0.01 (within each habitat). The vertical dashed line separates pigeons and doves (left) from other species (right).

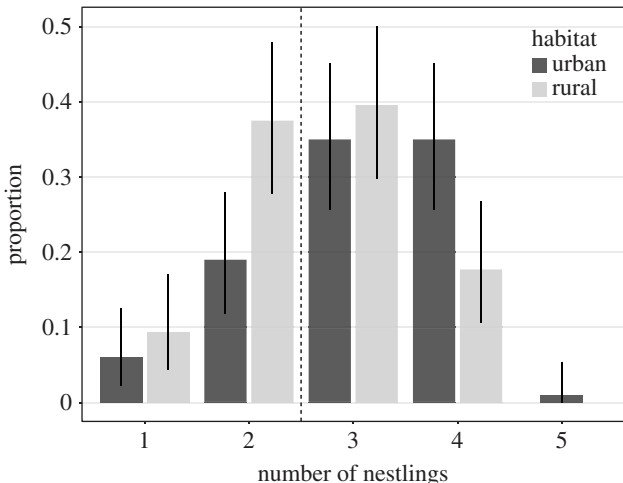

**Figure 6.** Proportion of nests containing 1, 2, 3, 4 or 5 nestlings within each habitat. Error bars show CI$_{95\%}$. The vertical dashed line separates small ($\leq$ 2 nestlings) from large broods ($\geq$ 3 nestlings).

that controlled for the effect of other variables (GLMM, $n = 195$: habitat: $\chi^2 = 0.006$, $p = 0.939$; temperature: $\chi^2 = 0.434$, $p = 0.543$). Nevertheless, laying date had a significant negative effect on the proportion of large broods ($\chi^2 = 6.65$, $p = 0.011$, figure 7): the earlier goshawk began laying, the larger the brood size was.

Because urban goshawks started breeding significantly earlier, it is possible that the larger brood size in the urban habitat is associated with this. To examine this possibility, we re-ran the previous model without laying date as regressor. This model indicated a significant effect of habitat type: the odds for goshawks to have large broods were 2.22 (CI$_{95\%}$ 0.984–4.73) times higher in the urban habitat than in the rural one (GLMM, $n = 195$: $\chi^2 = 5.54$, $p = 0.036$); temperature remained non-significant in this model ($\chi^2 = 0.295$, $p = 0.551$).

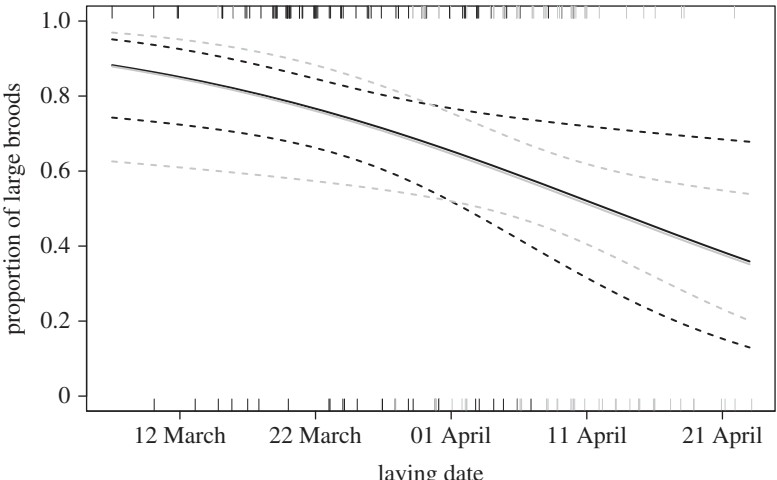

**Figure 7.** Predicted probability for a brood containing three nestlings or more as a function of laying date. See legend of figure 3 for further details. Habitat = urban (dark grey) and rural (light grey). Raw data are shown as jittered ticks along the *x*-axis at the bottom (small broods) or top (large broods) of the plot.

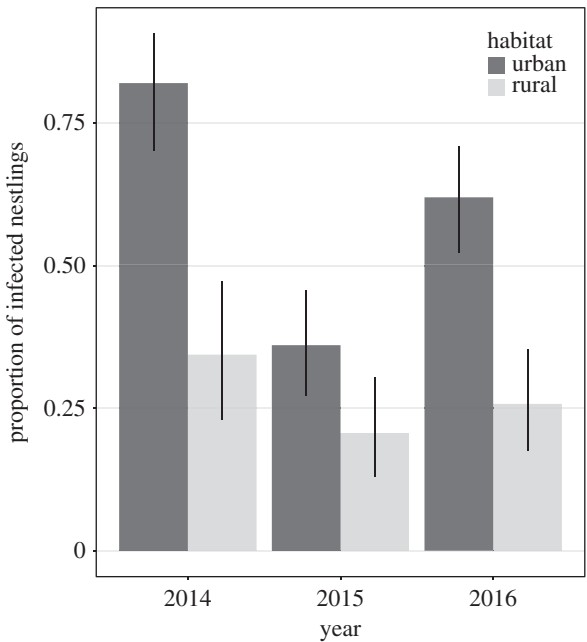

**Figure 8.** Proportion of nestlings (with $CI_{95\%}$) with *Trichomonas gallinae* infection in the three sampling years in urban and rural populations.

### 3.1.4. Health status

The chance of a goshawk nestling to be infected by *T. gallinae* was 2.83 ($CI_{95\%}$ 1.06–8.44) times higher in urban compared with rural habitats, although this effect was not significant (GLMM, $n = 542$: $\chi^2 = 4.90$, $p = 0.061$). Specifically, 160 out of 285 urban nestlings tested positive for the parasite (56.1%), whereas only 67 out of 257 rural nestlings did (26.1%). Prevalence varied significantly between sampling years ($\chi^2 = 19.8$, $p = 0.001$), being considerably lower in 2015 than in 2014 and 2016 (figure 8). Other independent variables did not significantly predict infection status (sex: $\chi^2 = 0.690$, $p = 0.397$; age: $\chi^2 = 0.508$, $p = 0.476$; laying date: $\chi^2 = 1.20$, $p = 0.326$; temperature: $\chi^2 = 0.483$, $p = 0.491$; number of nestlings: $\chi^2 = 0.079$, $p = 0.792$).

Eleven urban (3.9%) and six rural nestlings (2.3%) showed clinical signs of trichomonosis. After controlling for other covariates, the contrast between urban and rural individuals appeared significant, although this finding should be interpreted cautiously given the number of individuals showing

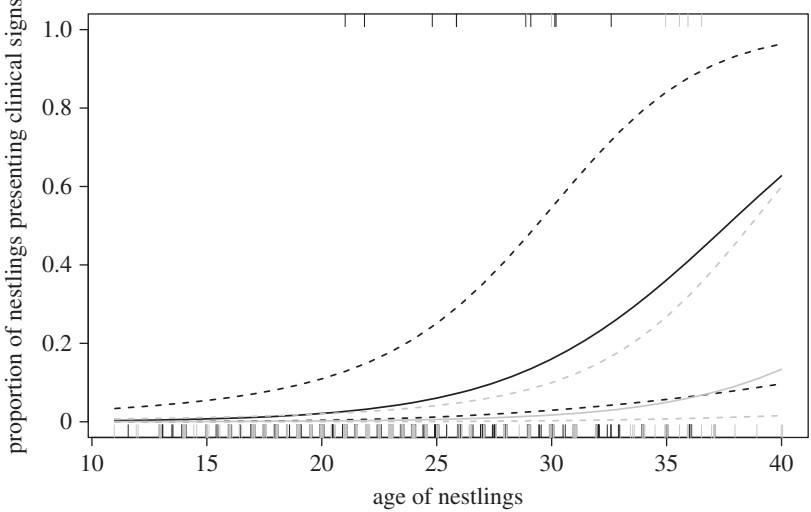

**Figure 9.** Predicted probability that goshawk nestlings showed clinical signs of an infection with *Trichomonas gallinae* as a function of age (in days). See legend of figure 3 for further details. Habitat = urban (dark grey) and rural (light grey). Raw data are shown as jittered ticks along the *x*-axis at the bottom (not infected) or top (infected) of the plot.

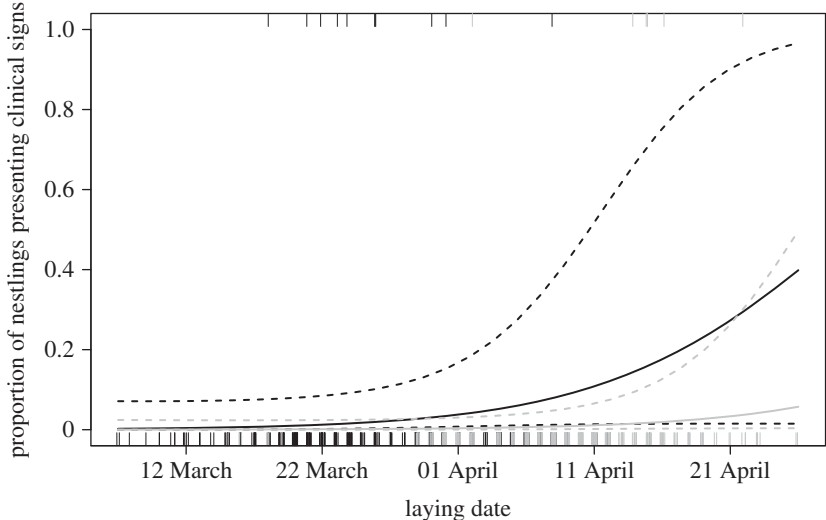

**Figure 10.** Predicted probability that goshawk nestlings showed clinical signs of an infection with *Trichomonas gallinae* as a function of laying date. See legend of figure 3 for further details. Habitat = urban (dark grey) and rural (light grey). Raw data are shown as jittered ticks along the *x*-axis at the bottom (no clinical signs) or top (clinical signs) of the plot.

clinical signs (GLMM, $n = 542$: $\chi^2 = 6.70$, $p = 0.016$). The chance to develop clinical signs appeared to increase with nestling age ($\chi^2 = 13.9$, $p = 0.001$, figure 9). The laying date of females showed a marginal effect ($\chi^2 = 3.78$, $p = 0.066$), with the chance of presenting clinical signs of the disease increasing the later the females initiated their broods (figure 10). Other independent variables had no significant effect (sex: $\chi^2 = 0.566$, $p = 0.473$; temperature: $\chi^2 = 0.790$, $p = 0.398$; number of nestlings: $\chi^2 = 1.58$, $p = 0.242$; year: $\chi^2 = 2.12$, $p = 0.376$).

### 3.1.5. Causes of mortality

The most common cause of mortality for urban goshawks was trauma as a result of a collision (figure 11): 33.1% (50 out of 151) of the cases were due to collisions with windows, 4.0% (6 out of 151) were due to collisions with vehicles, and 18.5% (28 out of 151) were due to collisions with unknown objects. In rural habitats, the most commonly recorded cause of death was also trauma: in 34.2% of cases, the circumstances remained unknown (13 out of 38), window strikes accounted for only 13.2% (5 out of 38) of deaths and vehicle collision for 7.9% (3 out of 38). The odds to die in a collision with windows

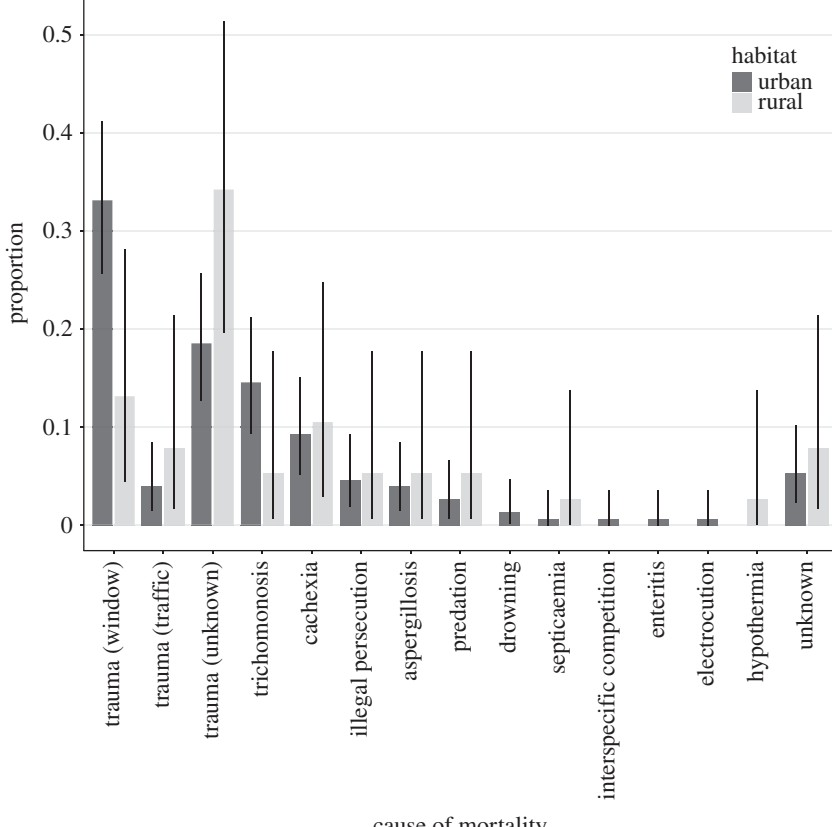

**Figure 11.** Proportion (with CI$_{95\%}$) of causes of mortality at urban and rural locations.

were 3.54 (CI$_{95\%}$ 1.39–10.9) times higher in the urban habitat than the rural one (GLM, $n = 186$: $\chi^2 = 7.35$, $p = 0.010$). Males were 2.31 (CI$_{95\%}$ 1.19–4.60) times more likely than females to die from striking a window ($\chi^2 = 6.23$, $p = 0.012$). The second most common cause of death in the urban habitat was trichomonosis, with 14.6% of all cases (22 out of 151), whereas cachexia (starvation) accounted for 10.5% (4 out of 38) of all deaths in rural habitats (figure 11). The odds that a goshawk died due to trichomonosis was 5.15 (CI$_{95\%}$ 1.08–43.7) times higher in the urban than the rural habitat (GLM, $n = 186$: $\chi^2 = 4.26$, $p = 0.042$). Age also had a significant effect ($\chi^2 = 33.5$, $p = 0.001$), with the risk of death due to the disease trichomonosis being highest for nestlings (8 out of 9), while sex was non-significant ($\chi^2 = 0.446$, $p = 0.502$).

## 4. Discussion

In this study, we compared five main characteristics of goshawks inhabiting either urban or rural habitats: their behavioural responses, diet composition, breeding performance, health status and causes of mortality. While our results indicate that urban life brings new challenges, a shift in diet, perhaps combined with behavioural changes, is likely to be the main reason why the 'phantom of the forest' may have become a successful citizen in several cities.

### 4.1. Behavioural responses

Behavioural flexibility can help animals adjust to human disturbance [17]. Our results indicate that urban and rural female goshawks behaved differently when we entered their territories as part of our research activities: urban goshawks were significantly more likely to produce alarm calls when their nest tree was climbed (and in a few cases, they even attacked the climber). Rural female goshawks could have been less likely to react to our presence for two reasons. First, if foraging conditions are poorer in rural habitats, females with older nestlings could have been forced to join their males' hunting efforts and were therefore less likely to be present during our visits. This scenario, however, was not strongly

supported by our data: the reaction rate of females decreased with the age of nestlings, and this decrease tended to be more marked in the rural than in the urban habitat as predicted, but none of these relationships were significant. Second, there could be a true behavioural difference between females from the two habitats—a scenario that is better supported by our data. Urban birds from other species have also been shown to be more aggressive than rural ones (e.g. Australian magpie [36,37]; noisy miner [38]). Behavioural adjustment with higher aggression and a higher stress-tolerance as part of the urbanization process could have allowed goshawks to breed in urban parks and cemeteries despite high levels of human disturbance [8–12,20,39]. However, it is currently unclear what the relative roles of genetic changes and phenotypic plasticity are in generating these behavioural differences between the two habitats.

Irrespective of habitat, we found that females that started laying eggs earlier were more likely to react to our intrusions. Møller & Nielsen [40] reported the same pattern for a rural population of goshawks in Denmark and suggested that early breeders could be of better phenotypic quality and could therefore afford to invest more time and energy into defensive behaviour.

## 4.2. Diet composition

Since food sources often differ between urban and rural habitats, animals must show some flexibility in their foraging behaviour to be able to colonize urban landscapes. We found that urban goshawks were more reliant on pigeons and doves than their rural counterparts (ca 65% for urban versus 35% for rural). This finding is consistent with the general notion that the goshawk is an opportunistic predator that preferentially hunts the most abundant and accessible prey of appropriate size [5,6,41–43] and also fits the observation that pigeons and doves are usually abundant in urban environments [44,45]. Other studies have shown that the brood size of goshawk pairs increases significantly with the proportion of pigeons in their diet [46,47] and an ability to kill selectively rare colour morphs may afford additional fitness benefits [47–49].

Strong reliance on pigeons, in turn, reduced species richness and diversity in urban goshawks' diet. This observation is again consistent with the idea that pigeons are high-quality prey for goshawks: Rutz & Bijlsma [50] found for a rural study population in The Netherlands that, with increasing levels of food shortage, the dominance of high-ranked prey species in the diet (both in terms of biomass and numbers) decreased and the number of small-bodied prey species in the diet increased. In most rural areas, goshawks seem to be unable to focus on one, or a few, profitable prey species, exhibiting relatively broader diets as a result, which may affect foraging effort, and ultimately reproductive output [51].

Overall, our diet analyses are in line with findings from earlier studies: a literature review by Rutz *et al.* [6] revealed that pigeons and doves made up 40.4% to 48.7% of the diet across urban goshawk populations, compared with just 3.3% to 34.1% in rural locations. This review, which included some of our study sites, observed that the proportion of pigeons and doves in goshawk diet appeared to have increased in recent years in urban areas, while it remained fairly stable in rural ones [6]. Consistent with this, we recorded more woodpigeons in the diet of urban goshawks compared with earlier studies. Dietary shifts have also been reported for other urban raptor species, like the common kestrel, which specializes in hunting birds instead of rodents [52,53]. Mammals are often found to exploit anthropogenic food sources in urban areas (red fox [54]; racoon [55]), which in racoons is associated with higher densities in urban areas [55] and higher birth rates [56].

## 4.3. Breeding performance

Several bird species have been reported to shift their *laying date* in response to living in urban environments, with some initiating clutches earlier (Cooper's hawk [15]; great tit [57]) and others later (burrowing owl [58]; common starling [59]) compared with non-urban areas. We found that urban goshawks started laying eggs on average 14 days earlier than their rural counterparts. This result is consistent with previous studies showing that urban goshawks initiate breeding about 10–14 days earlier [6], and that rural goshawks start breeding earlier where their habitat is more urbanized [60]. The two most likely reasons for the earlier onset of egg laying in urban areas are increased availability of prey, including year-around access to key species [6], as well as elevated temperatures [25]. Favourable weather conditions at the beginning of the breeding season may affect food availability and are known to be associated with early egg laying in goshawks [61,62] and many other bird species [63]. Our diet composition estimates are consistent with the first scenario. However, contrary to what has been found in other birds [64], we did not detect a significant effect of temperature.

Similarly, Looft [65] did not observe a change in laying date due to the warming climate in one of our rural goshawk study populations (Schleswig) that was monitored for over 50 years.

Among raptors, different types of breeding responses to urban living conditions have been documented [6,52,66]. For example, brood size is described to be larger in urban merlins [67], but smaller in urban common kestrels [52], and no clear pattern is described for sparrowhawks [68]. Furthermore, in mammals, like racoons, higher birth rates have been reported for urban areas compared with rural ones [55]. We found that urban goshawks exhibit slightly larger brood sizes than their rural counterparts. This is probably the result of favourable foraging conditions. Solonen *et al.* [69] reached the same conclusion, but they did not consider the potentially confounding effect of laying date. Interestingly, when we controlled for laying date in our statistical analyses, the difference in brood size between urban and rural habitats disappeared. As described for other raptor species (e.g. sparrowhawk [70]; merlin [67]; Cooper's hawk [15]), the increase in brood size we observed in goshawks seems directly related to an earlier onset of breeding. Krüger & Lindström [71] found that rural goshawk territories that were occupied earlier had a larger mean brood size and assumed that the quality of birds, rather than the habitat they occupied, might affect reproductive success. It is indeed possible that urban life attracts birds of higher phenotypic quality, as indicated by the behavioural differences we found and the earlier laying date in the urban habitat. Penteriani *et al.* [72] assumed that territory quality in combination with individual features may shape goshawks' foraging behaviour and therefore ultimately its breeding performance.

## 4.4. Health status

Few studies have explicitly examined disease threats to urban wildlife [14,15,73,74], although an association between breeding in urban habitats and disease transmission is often assumed *a priori* without being explicitly tested [2,63]. For urban goshawks, dense pigeon populations offer an excellent food supply, yet this prey is also the main host of the agent of trichomonosis [21], a disease known to affect survival and reproductive output in raptors [14,15,75]. Thus, pigeons potentially act as the main source of infection [73]. Accordingly, we recorded a higher proportion of infected nestlings in urban compared with rural populations (55.4% versus 25.9%). A link between urban lifestyle and infection risk is further supported by a small but significant difference in the percentage of cases where nestlings exhibited clinical signs of trichomonosis (3.9% versus 2.3%). An elevated prevalence of wildlife diseases in urban areas has also been found for some other bird species as well as other taxa [2].

## 4.5. Causes of mortality

Key threats to urban birds are accidental death due to collision with anthropogenic objects [24,76–78] and exposure to pathogens [14]. Across all birds collected and investigated in German cities, Stenkat *et al.* [23] identified trauma as the main cause of mortality, accounting for 62% of deaths, followed by parasitic diseases (18%). We found that a third of all recorded deaths for urban goshawks were due to collisions with glass structures. Urban goshawks were around three times more likely to die from window strikes compared with rural individuals. Hager [77] observed that windows affect *Accipiter* spp. hawks (which include goshawks) and falcons more than other raptors. The pursuit hunting style of goshawks makes them particularly susceptible to such accidents [5,39]. We estimated that the odds to strike a window were about twice as high for males compared with females. This sex difference probably reflects the fact that male goshawks provide their partners and chicks with food during the breeding season, associated with elevated hunting and activity levels [5]. Likewise, collisions with anthropogenic objects have been described as the main source of mortality in other taxa. In mammals, trauma due to collisions with vehicles is the most common source of mortality in suburban areas (racoons [55]; red foxes [79]). Pathogens seem to impact our urban study populations as well: we found that the second most common cause of mortality for urban goshawks was trichomonosis (14.6%). This parasitic infection was about five times more likely as a cause of death among urban birds compared with rural ones. The parasite is known to mainly infect nestlings, possibly due to a more favourable oral pH level [80]. Breeding losses due to trichomonosis have accordingly been proposed as a population limiting factor in Cooper's hawks [14] and in goshawks from Great Britain [81]. In mammals, higher mortality rates in cities due to diseases have also been described in several species. In racoons, the main source of mortality in urban areas is diseases [55], and red squirrels are believed to have disappeared from Norfolk, UK, due to higher infection rates in the city [82].

# 5. Conclusion

To our knowledge, our study is one of the most comprehensive investigations to date to compare the biology of a raptor species across urban and rural habitats, with standardized methodology and appropriate population-level replication [66]. We found that goshawks breeding in urban habitats seemed to be bolder than their rural counterparts and, benefitting from a favourable food supply, initiated breeding earlier and enjoyed comparatively larger brood sizes. Yet, breeding in urban environments comes with two notable costs: an elevated risk to become infected with the agent of trichomonosis and to collide with objects. Estimating mortality rates and lifetime reproductive success is challenging in long-lived species, as it requires following large numbers of individuals throughout their entire lives. Without detailed knowledge of the vital rates of urban and nearby rural populations, it is impossible to draw firm inferences about the suitability of these different habitats for goshawks. Nonetheless, in light of our findings and earlier work, and assuming that urban populations can sustain themselves without the constant need of immigrants from surrounding rural populations [6,10], we consider at least some cities to be high-quality breeding habitats for goshawks. This is particularly relevant in the context of a possible recent deterioration of rural habitats; several authors have raised concerns that prey shortages could have led to significant declines in rural goshawk populations [50,60]. Therefore, we propose to classify the goshawk an 'urban exploiter' under Blair's scheme [4]. Although the species does not depend on artificial resources like peregrine falcons [3], urban goshawks prefer old and high trees for nesting with forest areas and open land [83,84], which are usually found in cemeteries and parks. Such habitats thus need to be preserved to ensure successful cohabitation of goshawks and humans in cities like Berlin, Hamburg and Cologne, where goshawks are already resident, as well as in cities that remain to be colonized by this iconic raptor.

That urban living conditions can simultaneously be beneficial and detrimental is the result of the complex effects that the environment can have on different life-history traits. Interestingly, our own species (*Homo sapiens*) is also thought to have experienced improved fecundity and reduced survival with the formation of large settlements that became possible with the advent of agriculture [85,86].

Data accessibility. Data and relevant code for this research work are stored in GitHub: https://github.com/courtiol/accipiteR; and have been archived within the Zenodo repository: https://doi.org/10.5281/zenodo.4271624 [87].

Authors' contributions. M.M. and O.Kro. conceived the ideas; M.M., O.Kro. and C.R. designed the methodology; M.M., O.Kro., M.E., L.G., M.L., G.M., Y.H., R.Z., H.W., N.K., R.A., O.K., A.-K.M., N.C., V.L., T.G. and A.H. collected the data; M.M. and A.C. analysed the data and prepared the supplementary information; M.M. wrote the first draft of the manuscript; and M.M. and A.C. produced further versions of the manuscript, with significant contributions to the main text from O.Kro., C.R. and O.K. All authors commented on the drafts and gave final approval for publication.

Competing interests. The authors declare that they have no conflict of interest.

Funding. M.M. was part of the Graduate School IMPact-Vector funded by the Senate Competition Committee grant (SAW-2014-SGN-3) of the Leibniz Association. M.M. is also an associated doctoral student of the GRK2046 from the German Research Foundation (DFG). We are grateful for additional funding (Jagdabgabe) from the 'Stiftung Naturschutz Berlin' (J0056 & J0088), the 'Ministerium für ländliche Entwicklung, Umwelt und Landwirtschaft des Landes Brandenburg' (35-21340/7+5-51/16), the 'Behörde für Wirtschaft, Verkehr und Innovation der freien Hansestadt Hamburg' (title: 'Gesundheitsstatus und Ausbreitungsverhalten von Habichtnestlingen in Hamburg') and the 'Ministerium für Energiewende, Landwirtschaft, Umwelt und ländliche Räume des Landes Schleswig-Holstein' (V 542–42902/2016).

Acknowledgements. We thank Beate Ludescher, Dirk Stoewe, Jutta Mann, William Verpoort, Fred van Duijnhoven, Anna Hermsen, Kees Schreven, Alexander Detjen, Bernd Reuter, Hanna Prüter and Jose Maria Chapa Gonzalez for field assistance; the Small Animal Clinic of the Free University Berlin, especially Lesley Halter, Felix Lackmann and Kerstin Müller, for the supply of goshawk carcasses; Susanne Auls, Miriam Hahn, Katja Hagen, Jörg Schurath, Ariane Senske and Kirsi Blank for assistance with necropsies; Colin Vullioud for assistance with drawing the figures and François Rousset for assistance with the R package spaMM. The publication of this article was funded by the Open Access Fund of the Leibniz Association.

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
