## [Reviewer comments · Royal Society Open Science]

Review History

RSOS-201356.R0 (Original submission)

Review form: Reviewer 1

Is the manuscript scientifically sound in its present form?

Yes

Are the interpretations and conclusions justified by the results?

Yes

Is the language acceptable?

Yes

Do you have any ethical concerns with this paper?

No

Have you any concerns about statistical analyses in this paper?

Yes

Recommendation?

Accept with minor revision (please list in comments)

Comments to the Author(s)

I have reviewed the manuscript when submitted to Proc B and was happy to see that the authors reworked the manuscript so that conclusions reflect the data more accurately. I only have a few minor comments left:

Abstract - can you provide some error estimates around the numerical values you provide in your abstract (i.e. CIs, SE, SD)?

l17 - We measured a variety of behavioural and other variables for goshawk populations in three urban and four rural study sites

Doesn't sound right - use the original phrasing or other

How about: We measured a variety of behavioural, diet, reproductive, health and mortality parameters for goshawk populations...

l.22 - larger brood size - can you provide a value here?

l26 - goshawks seem to thrive

rephrase to: goshawks appear to thrive

Results

l194-195 - please compare comparable quantities - urban alarm calls vs rural alarm calls, and not urban alarm calls vs rural no calls

l209-211: can you double check the very low p-value from the Mann-Whitney test? It feels super high given the raw data

Decision letter (RSOS-201356.R0)

Dear Mrs Merling de Chapa

On behalf of the Editors, we are pleased to inform you that your Manuscript RSOS-201356 "Phantom of the forest or successful citizen? Analysing how Northern goshawks (*Accipiter gentilis*) cope with the urban environment" has been accepted for publication in Royal Society Open Science subject to minor revision in accordance with the referees' reports. Please find the referees' comments along with any feedback from the Editors below my signature.

Please submit your revised manuscript and required files (see below) no later than 7 days from today's (ie 09-Nov-2020) date. Note: the ScholarOne system will 'lock' if submission of the revision is attempted 7 or more days after the deadline. If you do not think you will be able to meet this deadline please contact the editorial office immediately.

Best regards,
Lianne Parkhouse
Royal Society Open Science
openscience@royalsociety.org

on behalf of Dr Kimberley Mathot (Associate Editor) and Kevin Padian (Subject Editor)
openscience@royalsociety.org

Associate Editor Comments to Author (Dr Kimberley Mathot):

Your manuscript has been re-reviewed by one of the referees from your original submission to Proc Roy Soc B. The reviewer is satisfied that this revised manuscript addressed the main concerns from the initial submission. However, they raise a number of points that I would like you to address before acceptance. These are all minor in nature and should be easy to address.

In addition, the data and scripts are currently available via GitHub. My understanding is that this is a dynamic storage system, meaning that contents remain editable. Please transfer the data and scripts to a static repository such as Dryad or FigShare and update your data accessibility statement.

Editorial Comments to Author:

As per the above, we ask that you please archive your GitHub code within the Zenodo repository: <https://guides.github.com/activities/citable-code/>. By doing this, a formal, citable DOI will be associated with your data record, and an open license (CC-BY preferred) can be applied to your data. We would then ask that you please update your data availability statement to read as:

"Data and relevant code for this research work are stored in GitHub: [GitHub URL here] and have been archived within the Zenodo repository: <https://doi.org/zenodo.....> [ref number]. At this stage, we ask that you please archive your GitHub code within the Zenodo repository: <https://guides.github.com/activities/citable-code/>. By doing this, a formal, citable DOI will be associated with your data record, and an open license (CC-BY preferred) can be applied to your data. We would then ask that you please update your data availability statement to read as:

"Data and relevant code for this research work are stored in GitHub: [GitHub URL here] and have been archived within the Zenodo repository: <https://doi.org/zenodo.....> [ref number].

Reviewer comments to Author:

Reviewer: 1

Comments to the Author(s)

I have reviewed the manuscript when submitted to Proc B and was happy to see that the authors reworked the manuscript so that conclusions reflect the data more accurately. I only have a few minor comments left:

Abstract - can you provide some error estimates around the numerical values you provide in your abstract (i.e. CIs, SE, SD)?

l17 - We measured a variety of behavioural and other variables for goshawk populations in three urban and four rural study sites

Doesn't sound right - use the original phrasing or other

How about: We measured a variety of behavioural, diet, reproductive, health and mortality parameters for goshawk populations...

l.22 - larger brood size - can you provide a value here?

l26 - goshawks seem to thrive

rephrase to: goshawks appear to thrive

Results

l194-195 - please compare comparable quantities - urban alarm calls vs rural alarm calls, and not urban alarm calls vs rural no calls

l209-211: can you double check the very low p-value from the Mann-Whitney test? It feels super high given the raw data

===PREPARING YOUR MANUSCRIPT===

===PREPARING YOUR REVISION IN SCHOLARONE===

<https://royalsociety.org/journals/authors/author-guidelines/#supplementary-material> to

include a suitable title and informative caption. An example of appropriate titling and captioning may be found at https://figshare.com/articles/Table_S2_from_Is_there_a_trade-off_between_peak_performance_and_performance_breadth_across_temperatures_for_aerobic_sc_ope_in_teleost_fishes_/3843624.

Author's Response to Decision Letter for (RSOS-201356.R0)

See Appendix A.

Decision letter (RSOS-201356.R1)

Dear Dr Merling de Chapa,

It is a pleasure to accept your manuscript entitled "Phantom of the forest or successful citizen? Analysing how Northern goshawks (*Accipiter gentilis*) cope with the urban environment" in its current form for publication in Royal Society Open Science. The comments of the reviewer(s) who reviewed your manuscript are included at the foot of this letter.

on behalf of Dr Kimberley Mathot (Associate Editor) and Kevin Padian (Subject Editor)
openscience@royalsociety.org

Associate Editor Comments to Author (Dr Kimberley Mathot):

Associate Editor

Comments to the Author:

Thank you for making the final revisions to your manuscript following the last round of reviews. Your manuscript has been accepted for publication.

Appendix A

Comments from the editors and reviewers:

We would like to thank the reviewer for his time and effort with our manuscript. Our responses to the reviewers' comments are listed below. For changes we used Microsoft Word's correction mode. Line numbers in the replies refer to the version in correction mode.

Editor comment:

As per the above, we ask that you please archive your GitHub code within the Zenodo repository: <https://guides.github.com/activities/citable-code/>. By doing this, a formal, citable DOI will be associated with your data record, and an open license (CC-BY preferred) can be applied to your data. We would then ask that you please update your data availability statement to read as:

"Data and relevant code for this research work are stored in GitHub: [GitHub URL here] and have been archived within the Zenodo repository: <https://doi.org/zenodo...> [ref number].

We did this and changed the Data availability statement accordingly (line 523-525)

Reviewer comments:

I have reviewed the manuscript when submitted to Proc B and was happy to see that the authors reworked the manuscript so that conclusions reflect the data more accurately. I only have a few minor comments left:

Comment 1:

Abstract - can you provide some error estimates around the numerical values you provide in your abstract (i.e. CIs, SE, SD)?

Author response:

We have now added information about the uncertainty of our estimates in the abstract (line 38, 40, 42, 43) and in the main text (line 238, 252-253, 272-273, 291, 295-296, 321-322, 323, 327). This led us to add details in the Method section too (line 210-215).

Comment 2:

l17 - We measured a variety of behavioural and other variables for goshawk populations in three urban and four rural study sites

Doesn't sound right – use the original phrasing or other

How about: We measured a variety of behavioural, diet, reproductive, health and mortality parameters for goshawk populations...

Author response:

We decided to use the original phrasing (line 34-35)

Comment 3:

l.22 - larger brood size - can you provide a value here?

Author response:

We now provided the odd ratio predicted by the model (line 43)

Comment 4:

l26 - goshawks seem to thrive

rephrase to: goshawks appear to thrive

Author response:

Rephrased according to the reviewers' suggestion (line 49)

Results**Comment 5:**

l194-195 - please compare comparable quantities - urban alarm calls vs rural alarm calls, and not urban alarm calls vs rural no calls

Author response:

We now changed it to the reviewers' suggestion and are now comparing urban alarm calls vs rural alarm calls and no calls for urban and rural goshawks (line 224-226)

Comment 6:

l209-211: can you double check the very low p-value from the Mann-Whitney test? It feels super high given the raw data

Author response:

We assume that the Reviewer meant the Mann-Whitney U test in line 246 in the original document and therefore checked this result. We can confirm that the p-value was correct (line 279).

Additional information:

In addition to the changes requested by the Editors and Reviewers, we adjusted some numbers in the sub-chapter *Health status* since we now include two additional individuals for which we retrieved data (Line 295-303). We have also recomputed all p-values in the text with the latest version of the R package spaMM for greater reliability. Our results remain unchanged by these two main modifications beyond changes in decimals.